# Peer review of "Ethical Implications of Chatbot Utilization in Nephrology"

_jpm, 2023, doi:10.3390/jpm13091363_

Round 1
Reviewer 1 Report
This paper looked at Ethical Implications of Chatbot Utilization in Nephrology, it is well written and provided good suggestions and examples for improvement of Chatbot. It is appropriate for publication.
Author Response
Reviewer 1
This paper looked at Ethical Implications of Chatbot Utilization in Nephrology, it is well written and provided good suggestions and examples for improvement of Chatbot. It is appropriate for publication.
Response: Thank you for taking the time to review our paper titled "Ethical Implications of Chatbot Utilization in Nephrology." We are pleased to learn that you found the paper well-written and that the suggestions and examples provided were beneficial. We deeply value your positive feedback concerning the overall organization and content of the paper. It is encouraging to know that our efforts to shed light on the ethical aspects of chatbot use in nephrology have been recognized.

Reviewer 2 Report
Authors have reviewed potential ethical implications of utilisation of chatbots in Nephrology. The review is comprehensive. I have few suggestions. A brief note on present status of chatbot use in Medicine and Nephrology could be added. Since this has major implications for patient management a note on whether clinical trials are required to check utility of chatbots, ethical issues in conduct of such trials and briefly about conducting randomised trials with blinding. Authors mention supervision and updating of use of chatbots. A note on who would be qualified as supervisors for this task would be helpful. Again points on the level of evidence and how updated the chatbot is and its presentation to patients in plain language needs to be mentioned. Also the ethical implication of potential for chatbot doing multitasking in terms of diagnosis followed by drawing up a list of therapies then suggesting and possibly sourcing the therapy for the patient needs exploration as technology for this already exists in other areas.
Overall the review looks complete.
Author Response
Reviewer 2
Authors have reviewed potential ethical implications of utilisation of chatbots in Nephrology. The review is comprehensive. I have few suggestions. A brief note on present status of chatbot use in Medicine and Nephrology could be added. Since this has major implications for patient management a note on whether clinical trials are required to check utility of chatbots, ethical issues in conduct of such trials and briefly about conducting randomised trials with blinding. Authors mention supervision and updating of use of chatbots. A note on who would be qualified as supervisors for this task would be helpful. Again points on the level of evidence and how updated the chatbot is and its presentation to patients in plain language needs to be mentioned. Also the ethical implication of potential for chatbot doing multitasking in terms of diagnosis followed by drawing up a list of therapies then suggesting and possibly sourcing the therapy for the patient needs exploration as technology for this already exists in other areas.
Overall the review looks complete.
Responses: Thank you for your meticulous review of our paper on the "Ethical Implications of Chatbot Utilization in Nephrology." We deeply appreciate your constructive feedback and are committed to integrating your valuable suggestions to enhance the quality of our paper.
Status of Chatbot Use in Medicine and Nephrology: We concur with your suggestion about including a section detailing the current status of chatbot use in both the broader field of Medicine and the specific domain of Nephrology. This will provide readers with a clearer context and establish the relevance of our discussion on ethical implications. The following text has been added in the revised manuscript:
“1.2 Status of Chatbot Use in Medicine and Nephrology
In the realm of medicine, chatbots have risen as an instrumental tool, seamlessly enhancing patient interactions, streamlining administrative workflows, and elevating healthcare service quality. Their multifaceted applications span from enlightening patients, sending medication alerts, assisting with preliminary diagnoses, to operational responsibilities like scheduling visits and gathering patient insights. Their ascendancy is attributed to a mix of elements, such as the widespread use of intelligent devices, amplified internet connectivity, and the consistent advancements in AI technologies.
Focusing on nephrology, chatbots furnish numerous advantageous contributions. Acting as digital aides for nephrologists, they present on-the-fly data evaluations of patient metrics, support patient inquiries, and even contribute to regular check-ins. Their value becomes pronounced in supporting patients with persistent kidney ailments, where regular oversight and swift communication significantly uplift patient well-being. Nonetheless, the integration of such technologies in vital areas does present its unique set of challenges and moral dilemmas.
Some of the leading chatbots include:
- ChatGPT, an initiative by OpenAI, is underpinned by the GPT (Generative Pre-trained Transformer) suite of language models, known for emulating human text creation. Embracing models such as GPT-4, ChatGPT stands out in crafting quality content, language translations, and delivering in-depth query responses.
- Bard AI, from the house of Google AI, taps into sophisticated language models like the Pathways Language Model 2 (PaLM 2) and its predecessor, the Language Model for Dialogue Applications (LaMDA). This allows Bard AI to decipher a vast range of prompts, inclusive of those that require logical, commonsensical, and mathematical insights.
- Bing Chat, a Microsoft brainchild, offers generalized information and insights across diverse subjects, including the medical realm. Bing Chat harnesses state-of-the-art natural language processing and AI-driven learning to emulate human conversational behaviors.
- Claude AI, birthed by Anthropic, operates on an exclusive language model termed Constitutional AI. Crafted with the principles of usefulness, safety, and integrity through its Constitutional AI technique, Claude excels in decoding intricate queries and delivering accurate, detailed answers.”
Clinical Trials and Ethical Concerns: Your point about the necessity of clinical trials to assess the utility of chatbots is well taken. We additionally included a segment detailing:
- The importance of clinical trials in this context.
- Ethical challenges associated with these trials.
- Considerations around conducting randomized trials with blinding for chatbots. The following text has been added in the revised manuscript:
“2.4. Clinical Trials and Ethical Concerns in Chatbot Utilization
The melding of technology with healthcare has positioned chatbots at the forefront of patient management discussions. Their advantages, along with the inherent complexities they bring, emphasize the need for thorough evaluation. Their role is not just a reflection of tech progress but plays a significant part in shaping patient health and safety outcomes. It becomes imperative to assess their accuracy, reliability, and effectiveness, viewing them not merely as software but as vital components in healthcare delivery. Clinical assessments provide the ideal avenue to gauge their performance in diverse medical settings.
Incorporating chatbots in such assessments reveals certain ethical questions. It is essential to guarantee that patients recognize their digital interlocutor and grasp the potential ramifications. Ensuring the protection of sensitive health records is crucial. There's also concern about chatbots mirroring or exacerbating biases present in their training data, which may skew treatment advice. Clearly delineating responsibility in cases where chatbots falter or malfunction becomes essential.
The notion of randomized trials with chatbots, where patients are uncertain if guidance comes from AI or a human, presents unique complexities. Such blind tests aim to directly attribute results to the chatbot, devoid of any patient preconceptions. One conceivable method could be employing a consistent interface for every interaction, obscuring the source of the advice. Yet, the ethical and logistical aspects of such an approach demand thorough examination.”
Supervision of Chatbots: The matter of who qualifies as a supervisor for chatbot utilization is indeed crucial. We additionally elaborated on:
Potential qualifications and credentials for such supervisors.
The importance of domain expertise combined with technical know-how.
Potential frameworks or guidelines for supervision. The following text has been added in the revised manuscript:
“Oversight of Digital Assistants in Nephrology
In the expanding domain of digital solutions in nephrology, the oversight of chatbots stands as a crucial matter. Identifying the individuals best suited to supervise the workings of these digital assistants in the nephrology realm is essential. Key aspects to consider include:
Required Expertise and Accreditation: The individual responsible for overseeing a chatbot, especially one that focuses on renal care, should possess not only relevant nephrological qualifications but also a comprehensive understanding of the chatbot's focus area. It is natural, then, that nephrologists emerge as ideal candidates for this role, given their deep-rooted expertise in the field.
Combination of Clinical and Technical Insight: Beyond the realm of nephrology, the individual supervising should also have a grasp on the technological nuances of the chatbot. It's this blend of clinical and tech insights that ensures the chatbot aligns with both medical standards and technological efficiency.
Operational Protocols and Best Practices: Implementing a well-defined operational blueprint for overseeing chatbots in nephrology is of utmost importance. Components of this blueprint might encompass regular assessments of the chatbot, fostering communication channels with its developers, and ensuring that the chatbot aligns with current nephrological standards and practices.”
Evidence Level and Presentation to Patients: We acknowledge the significance of clearly communicating the evidence level behind chatbot recommendations. In addition:
- We additionally discussed the importance of keeping chatbots updated with the latest medical knowledge.
- We additionally emphasized the ethical need to present information to patients in plain language, ensuring transparency and comprehension. The following text has been added in the revised manuscript:
“Evidence Level and Presentation to Patients
In the rapidly evolving landscape of nephrology, chatbots, as digital tools, bear a significant responsibility when delivering diagnostic or treatment recommendations. This necessitates clarity in the levels of evidence underpinning such advice.
Understanding Evidence Levels: It is of utmost importance for patients and clinicians alike to have an understanding of the evidence strength behind the recommendations provided by the chatbot. Differentiating between a suggestion derived from a high-quality, multi-center randomized control trial and one based on less robust studies or anecdotal evidence can have profound implications for treatment pathways and patient outcomes. Thus, there should be a clear and structured way for chatbots to communicate this differentiation.
Continual Updates with Latest Medical Knowledge: As with any clinical tool, stag-nation equates to obsolescence. Chatbots must be dynamic entities, regularly assimilating the latest in medical research and best practices. The pace at which new findings emerge in nephrology underscores the importance of this continual learning process. This not only ensures the relevance of the chatbot but also upholds its reliability, a factor that could determine its broader acceptance in the medical community.
Ethical Imperative of Transparent Communication: While accuracy is essential, the manner in which information is conveyed to patients is equally critical. It is an ethical imperative to ensure that information, especially medical, is presented in a manner that is both transparent and easily comprehensible to patients. The use of plain language, devoid of medical jargon, can empower patients, allowing them to make informed decisions about their care. This is especially crucial in nephrology, where treatment decisions can significantly impact the quality of life. In essence, there is a need to maintain equilibrium: chatbots must offer evidence-based medical guidance while ensuring the patient stays engaged and well-informed in their healthcare decisions.”
Your feedback has provided a comprehensive roadmap for refining our paper to ensure it is both accurate and thorough. We are grateful for the time and expertise you dedicated to reviewing our work and for highlighting areas of potential improvement. We are confident that, with these revisions, the paper will provide a more in-depth and nuanced examination of the topic at hand.

Reviewer 3 Report
Thank you for providing us a detailed example of how it's going with using AI in medicine. The authors have described the ethical utilization of chatbot in clinical medicine. The authors have concluded that its use must accord high priority to ethical considerations to guarantee patient privacy, fairness, and informed decision-making.
The authors have provided examples for manipulative behavior of AI, Lack of Human Oversight, over-simplifying of decisions, patient privacy concerns, language and knowledge gap between chatbot and human, and lack of transparency.
Although I am interested to know all about the use of such tools from the patient's side, the authors have neglected the other part "from the research/healthcare side". I suggest that the authors consider putting examples on how the situation from a healthcare provider side of view.
I do appreciate Table (1), illustrating the concerns and the recommendation for them.
The manuscript is very easy to read and understand, very well written and assess a global concern of ethical consideration of new AI tools.
Author Response
Reviewer 3
Thank you for providing us a detailed example of how it's going with using AI in medicine. The authors have described the ethical utilization of chatbot in clinical medicine. The authors have concluded that its use must accord high priority to ethical considerations to guarantee patient privacy, fairness, and informed decision-making.
The authors have provided examples for manipulative behavior of AI, Lack of Human Oversight, over-simplifying of decisions, patient privacy concerns, language and knowledge gap between chatbot and human, and lack of transparency.
Although I am interested to know all about the use of such tools from the patient's side, the authors have neglected the other part "from the research/healthcare side". I suggest that the authors consider putting examples on how the situation from a healthcare provider side of view.
Responses: We sincerely thank you for your thoughtful and constructive feedback on our manuscript. We are grateful for your appreciation of the clarity and importance of our work, as well as for pointing out areas of improvement. Addressing your comments, we have prepared the following responses and modifications to include "from the research/healthcare side" as suggested.
“Ethical Challenges of Chatbot Integration in Nephrology Research and Practice
The rapid integration of chatbots and artificial intelligence (AI) tools within the healthcare sector offers both emerging opportunities and significant challenges, especially when analyzed from the perspective of research and healthcare professionals. Within the realm of nephrology, which necessitates a comprehensive understanding and expertise, the ill-advised application of AI has the potential to result in unexpected and potentially ad-verse outcomes.
A current concern is the prospective utilization of chatbots by medical trainees for intricate test responses. Although these AI systems may not have been initially designed for such tasks [77], the temptation to employ them for expedient solutions could hinder authentic learning, foster excessive dependence, and potentially compromise patient outcomes in real-world settings. This apprehension extends beyond simple ethical considerations, suggesting a potential decline in professional competencies when these technologies are adopted without critical assessment. Moreover, the incorporation of chatbots in scholarly research has elicited multiple concerns. There have been documented cases where chat-bots have been employed to compose manuscripts and scholarly articles. While this might expedite the composition process, an excessive dependence without rigorous review can result in errors or potential misinterpretations. Of significant concern is the potential employment of chatbots to produce references for scholarly publications [78]. The capability of AI to either intentionally or unintentionally produce erroneous references or introduce mistakes poses a threat to the integrity of medical literature [78]. Such practices not only violate the principles of academic honesty but also pose the risk of disseminating inaccurate or deceptive information, which could profoundly impact patient care and the broad-er understanding of science. As chatbots and AI tools increasingly influence nephrology and other medical fields, it becomes imperative to rigorously assess their ethical application, particularly in research and healthcare domains. The aim should be to seamlessly integrate innovation with ethical principles, ensuring that these tools augment, rather than undermine, the credibility and expertise of the medical profession. It is noteworthy that there has been a discernible increase in policy development by academic journals concerning the recognition and inclusion of AI in manuscript preparation. However, this acknowledgment is accompanied by a stipulation. As AI becomes an integral component of scholarly writing, it is essential to maintain vigilance and consistently monitor AI contributions, safeguarding the precision and veracity of scientific inquiry.”
Figure 9. Ethical Challenges of Chatbot Integration in Research and Practice
I do appreciate Table (1), illustrating the concerns and the recommendation for them.
Responses: We are glad you found Table 1 illustrative and informative. We believe that such tables are pivotal in summarizing and providing clear recommendations for the concerns raised, and we are pleased it resonated with your perspective.
The manuscript is very easy to read and understand, very well written and assess a global concern of ethical consideration of new AI tools.
Responses: Your positive feedback on the readability and global relevance of our manuscript is highly motivating. We aimed to provide a comprehensive yet comprehensible overview of the topic, and we are pleased to know we achieved that.
Thank you once again for your time and insights. We value and respect your expertise, and we are confident that your suggestions will greatly improve the quality and relevance of our work.

Round 2
Reviewer 3 Report
The manuscript is accepted at its present form.